# Combinatorial Treatment of Tinzaparin and Chemotherapy Can Induce a Significant Antitumor Effect in Pancreatic Cancer

**DOI:** 10.3390/ijms22137053

**Published:** 2021-06-30

**Authors:** Panagiotis Sarantis, Alexandros Bokas, Adriana Papadimitropoulou, Evangelos Koustas, Stamatios Theocharis, Pavlos Papakotoulas, Dimitrios Schizas, Alexandros Papalampros, Evangelos Felekouras, Athanasios G. Papavassiliou, Michalis V. Karamouzis

**Affiliations:** 1Molecular Oncology Unit, Department of Biological Chemistry, Medical School, National and Kapodistrian University of Athens, 11527 Athens, Greece; panayotissarantis@gmail.com (P.S.); alexanderbokas@outlook.com (A.B.); vang.koustas@gmail.com (E.K.); 2Department of Medical Oncology, ‘Theageneio’ Cancer Hospital, 54639 Thessaloniki, Greece; papakotoulas@gmail.com; 3Center for Basic Research, Biomedical Research Foundation of the Academy of Athens, 11527 Athens, Greece; adapapadim@gmail.com; 4First Department of Pathology, Medical School, National and Kapodistrian University of Athens, 11527 Athens, Greece; statheocharis@yahoo.com; 5First Department of Surgery, Medical School, National and Kapodistrian University of Athens, 11527 Athens, Greece; schizasad@gmail.com (D.S.); a_papalampros@hotmail.com (A.P.); evangelosfelek@hotmail.com (E.F.)

**Keywords:** pancreatic cancer, tinzaparin, chemotherapy, nab-paclitaxel, gemcitabine, xenografts

## Abstract

Pancreatic Cancer (PC) is recognized as a highly thrombogenic tumor; thus, low-molecular-weight heparin (LMWH) such as tinzaparin is routinely used for PC patients. On the basis of combinatorial therapy approaches to treat highly malignant and refractory cancers such as PC, we hypothesized that tinzaparin can augment the effectiveness of traditional chemotherapeutic drugs and induce efficient antitumor activity. PANC-1 and MIAPaCa-2 were incubated alone or in combination with tinzaparin, nab-paclitaxel and gemcitabine. In vivo evaluation of these compounds was performed in a NOD/SCID mouse using a model injected with PANC-1. Tinzaparin enhances the anti-tumor effects of nab-paclitaxel and gemcitabine in mtKRAS PC cell lines via apoptosis in in vitro experiments. The triple combination power acts through the induction of apoptosis, reduction of the proliferative potential and angiogenesis; hence, contributing to a decrease in tumor volume observed in vivo. The triple regimen provided an extra 24.3% tumor reduction compared to the double combination (gemcitabine plus nab-paclitaxel). Combinatorial strategies can create novel therapeutic approaches for the treatment of patients with PC, achieving a better clinical outcome and prolonged survival. Further prospective randomized research is needed and the investigation of various concentrations of tinzaparin above 150 UI/Kg, would potentially provide a valuable synergistic effect to the conventional therapeutic compounds.

## 1. Introduction

Pancreatic cancer (PC) is the fourth leading cause of cancer deaths, with a five-year survival rate of 9% [1]. The two main types of cancer are adenocarcinoma (accounting for about 85% of cases) and pancreatic endocrine tumors (which make up less than 5% of all cases). The incidence of pancreatic cancer has been steadily increasing in recent years and is predicted to become the second leading cause of death in Western countries by 2030 [2,3]. Despite extensive research performed, prognosis and therapeutic strategies have only slightly improved, mainly due to late diagnosis and limited responsiveness to potential compounds [4]. Causes of poor survival are considered to be the low effectiveness of adjuvant chemotherapy after surgery, the undetectable micro-metastases and the development of drug resistance [5]. PC patients show a rapid progression of the disease and few of them survive more than a year. Even for patients undergoing radical surgery, the median survival remains low at approximately 18 months. Thus, despite advances in understanding the basic biology of the disease, survival rates have remained substantially unchanged over the past 30 years [6].

Gemcitabine is currently the standard treatment option for advanced and metastatic PC and other types of cancers such as bladder and breast. It was initially applied as an antiviral agent and was later widely used as an anti-cancer chemotherapeutic agent for various solid tumors and lymphomas [7]. Despite its effectiveness as a drug in a significant number of cases, the different response rates, increased toxicity and development of resistance in some patients, remain important causes of ineffectiveness in recurrent tumors [8]. In order to compensate for these limitations, a high dose of gemcitabine (approximately 1000 mg/m^2^) is usually administered. [9].

The combination of gemcitabine with nab-paclitaxel has shown higher rates of response and increased survival in patients with pancreatic cancer. Nab-paclitaxel is a colloidal suspension of particles homogenized in human serum albumin-bound to paclitaxel. In mouse models, the combined treatment of gemcitabine and nab-paclitaxel resulted in increased concentrations of gemcitabine in plasma and tumors [10]. Nab-paclitaxel is associated with the albumin receptor gp60 as well as with the secreted acid-rich and cysteine-rich SPARC protein, which is expressed in many cancer cells [11,12,13]. However, the combination of the two chemotherapeutic drugs, nab-paclitaxel and gemcitabine, has been shown to augment hematological toxicity compared to monotherapy [14].

Tinzaparin, along with dalteparin and enoxaparin, belongs to the low molecular weight heparins (LMWH), which are derivatives of unfractionated heparin (UFH), through depolymerization of heparin by heparinase enzyme [15]. LMWHs do not inhibit thrombin, but they can attenuate coagulation factor Xa [16,17]. They offer a significant therapeutic advantage, having a more predictable dose-response to anticoagulant therapy, improved subcutaneous bioavailability, dose-dependent clearance, longer half-life, lower incidence of thrombocytopenia, and reduced requirement for laboratory monitoring [18].

Heparin administration has been extensively evaluated as a treatment for syndromes such as acute coronary heart disease, deep vein thrombosis (DVT) and pulmonary embolism (PE). It has also been studied for the prevention of venous thromboembolism (VTE) in several high-mobility patients [19]. In recent years, LMWH has successfully replaced UFH in both thromboprophylaxis and initial treatment of VTE. Studies on the efficacy and safety of tinzaparin in DVT have concluded that it constitutes an effective therapeutic approach for the secondary prevention of thromboembolic events [7,20].

PC is recognized as a highly thrombogenic tumor, with over 20% of patients suffering from venous thromboembolism during the disease [21,22]. Chemotherapeutic agents are associated with thrombotic mechanisms, including the release of coagulants and cytokines from tumor cells and the production of toxic agents that act directly on the endothelium. Surgery, the first-line treatment for many cancer patients, is known to activate the hemostatic system, and in this way, cancer treatments activate a coagulation cascade, promoting thrombosis and tumor growth [23].

Beyond its role in reducing VTE occurrence in pancreatic cancer patients, tinzaparin may help to prolong survival by affecting tumor progression, metastasis formation and angiogenesis based on in vitro studies and in vivo studies on breast cancer [24,25]. This study aimed to determine the role of tinzaparin in pancreatic cancer. More specifically, we intended to decipher whether it contributes to tumor shrinkage, especially after co-administration with gemcitabine and nab-paclitaxel and identify possible in vitro and in vivo mechanisms by which this inhibition is achieved.

## 2. Results

### 2.1. Tinzaparin Enhances the Anti-Tumor Effect of Nab-Paclitaxel and Gemcitabine in mtKRAS PC Cell Lines

In our study, we examined the effect of co-treatment with nab-paclitaxel (Nab-P), gemcitabine (G) and tinzaparin (T) in PC PANC-1 and MIAPaCa-2 cell lines. Both cell lines were exposed to 1 μM Nab-P, 1 μM G, 2 UI/mL T alone or in combination for 48 h; after exposure, cell viability was quantified by the XTT-viability assay. Cells without any drug administration were used as control. In the PANC-1 cell line, tinzaparin decreased cell viability by 20% whereas the combination of T + G and T + Nab-P decreased cell viability by 41% and 42.5% respectively. In the triple combinatorial scheme (Nab-P + T + G), cell viability of PANC-1 was decreased by 55% after 48 h with statistical significance *p* = 0.0006. Compared to T + G, the *p*-value was 0.0337 and compared to T + Nab-P, the *p*-value was 0.0147.

In, MIAPaCa-2 cell line, Nab-P reduced cell viability by 21.5% and the combination of T + Nab-P and T + G lead to an additional reduction of 8.5%. The triple regimen T + G + Nab-P resulted in a significant decrease of the total cell viability after 48 h by 40% compared to the untreated control (*p* = 0.002). Compared to T + G, *p* = 0.0658 and compared to T + Nab-P, *p* = 0.0319.

It appears that the triple combinatorial scheme of T + G+ Nab-P provides an extra reduction of cell viability in both cell lines than the agents alone (Figure 1).

### 2.2. Stimulation of Apoptotic Pathways Results to Decreased Cell Survival

In order to further explore the mechanisms by which the triple combination of T, G and Nab-P affects the viability of PC cell lines PANC-1 and MIAPaCa-2, we performed Western blot assays for specific apoptotic markers. More specifically, cells were exposed to 1 μM Nab-P, 1 μΜ G, 2 UI/mL T alone or in combination for 48 h and cell lysates were quantified and used to detect two specific apoptotic markers, PARP-1 and caspase-3. The double combination of T + G or T + Nab-P slightly increased the cleavage of caspase-3 and PARP in both cell lines. Triple combinatorial scheme of T + G + Nab-P in PANC1 and MIAPACA2 resulted in apoptotic cell death. The presence of apoptotic cell death was confirmed by the detection of PARP-1 cleavage and cleaved caspase-3 (Figure 2). Our results highlight the vigorous antitumor activity of tinzaparin when combined with Nab-paclitaxel and gemcitabine in in vitro experiments in PC models according to the mutant KRAS profile.

### 2.3. In Vivo Experiments with N0D/SCID Mouse Model Injected Heterotopically with Pancreatic Cancer Cells, Shows That Tinzaparin Administration Acts Synergistically with Chemotherapeutic Drugs and Provides a Significant Tumor Reduction

The results of our in vitro experiments, support the hypothesis that combinatorial treatment with tinzaparin, gemcitabine and nab-paclitaxel decrease the viability of PC cell lines. In light of these outcomes, we tried to evaluate our in vitro results in xenograft mouse models. For xenograft studies, we used the PANC-1 PC cell line, bearing mutant KRAS, because it showed more reliable and significant results in our in vitro experiments. The combinatorial treatment effectively decreased tumor volume as shown in Figure 3.

It is noteworthy that tumor reduction observed in mice receiving the triple combination chemotherapy was 74.5% compared to the control (*p* < 0.0001). The inclusion of tinzaparin in the triple regimen provided an additional 8.2% and 24.3% tumor reduction compared to the control and the double combination (gemcitabine plus nab-paclitaxel) respectively (*p*-value for the double combination was 0.0329). A similar trend was observed after administration of nab-paclitaxel plus tinzaparin compared to control (extra 6.3% reduction) and when compared to nab-paclitaxel alone (extra tumor reduction approximately 16.1%). The combination of tinzaparin with gemcitabine also led to a reduction of 11.8% compared to the control and 13.9% compared to gemcitabine alone (Figure 3).

### 2.4. Immunofluorescence with PCNA Shows That the Triple Combination Has Significant Reduction in Cell Proliferation in Mouse Xenografts

Next, we performed immunofluorescence to the extracted tumors in order to examine cell proliferation status with the use of a PCNA marker. Tumors in untreated mice had -as expected- increased proliferation potential (76.25% ± 5.68% positive cells) but when treated with the triple combination, tumors showed a 3.86-fold less proliferative capacity (*p* < 0.0001). When triple combination compared to Νab-P + G there was a reduction of proliferating cells by 30.01% (*p* < 0.0001) (Figure 4A,B).

### 2.5. Stimulation of Apoptotic Pathways Is One of the Mechanisms Leading to Tumor Reduction In Vivo

Extracted tumors were further analyzed with immunohistochemistry using an antibody against caspase-3. The percentage of apoptotic cells in the triple combination was 24.83% ± 3.63%, while in the control it was 2.75% ± 1.84%, a significant increase of 9.03-fold (*p* < 0.0001). A similar substantial increase of apoptosis was observed (24.89%, *p* = 0.0007) when the triple combination is compared to the double, revealing a substantial synergistic effect of tinzaparin (Figure 5A,B).

### 2.6. Triple Drug Combination Negatively Affects Tumor Neoangiogenesis

Finally, an important factor for tumor growth is the formation of new vessels that would provide all essential nutrients for its development. The effect of drug treatments was investigated by immunohistochemistry with the use of anti-VEGFR2. As pointed out in Figure 6, there was a large number of vessels formatted in the control sample, but this phenomenon was reversed with the use of tinzaparin, especially in the triple combination where less than three vessels were counted at each field of measurement.

## 3. Discussion

Pancreatic cancer constitutes one of the most lethal diseases, with an average 5-year survival rate of less than 10% [1]. About 60–80% of patients are diagnosed with advanced cancer as it invades the surrounding tissues (locally advanced) or has disseminated outside the pancreas (metastatic) [26]. KRAS mutations comprise the most frequent alterations observed in pancreatic ductal adenocarcinoma (PDAC), therefore this type of cancer is considered one of the most RAS-dependent of all cancers [27,28]. As the disease shows a very high mortality rate, it is imperative to discover novel and more effective treatments, especially for those carrying the KRAS mutation.

The presence of desmoplasia is a feature of the pathogenesis and evolution of PC that represents 60–90% of the total tumor mass [29] and can be recognized in both primary and metastatic areas of the tumor [30]. Fibroblasts are the predominant cell type within the TME, where they form a heterogeneous group of cells that contribute to ECM formation and support tumor growth [31]. Pancreatic stellate cells represent a subset of cancer-related fibroblasts present in a dormant state in healthy pancreatic tissue. Once activated, i.e., as a result of inflammation, injury, or tumor formation, they adopt a myofibroblast-type phenotype with high proliferative capacity and increased extracellular matrix protein secretion [32]. TME exerts various actions ranging from tumor formation, cancer spread and resistance to treatment [33]. TME is further characterized by severe hypoxia and, when combined with vascular compression induced by desmoplasia, it triggers the process of angiogenesis to support the tumor’s constant need for nutrients [34] Conventional drugs, such as gemcitabine, are not able to penetrate the rich and thick layer of TME and therefore lead to drug tolerance.

In our study, the PC line PANC-1 was selected, among other reasons, because this type of cell exhibits stromal characteristics and provides an adequate simulation of real conditions of pancreatic cancer concerning the TME [35,36]. LMWHs can reduce fibrin formation and contribute to ECM degradation and thus result in TME weakening. It could be hypothesized that when combined with drugs such as nab-paclitaxel and gemcitabine, LMWHs act synergistically to help these agents in penetrating the TME and reaching cancer cells more efficiently in order to eliminate them, which is reflected in our in vivo results.

Tumors release vascular growth factors, such as VEGF and basic fibroblast growth factor (bFGF), which, along with other cytokines, stimulate angiogenesis by interacting with their high-affinity intracellular activity receptors. In humans, therapeutic doses of UFH actually cause increased levels of growth factors, such as plasma bFGF [18]. Heparin affects the activity of other factors involved in angiogenesis and tumor growth, in addition to VEGF and bFGF [37]. Growth factor TGF is a potent immunosuppressive and essential regulator of growth, differentiation and adhesion of various cells. It is expressed in cancer cells and its overproduction is associated with an unfavorable prognosis.

LMWHs, on the other hand, can inhibit the binding of growth factors to their receptors while they are able to negatively affect the rate of angiogenesis [38]. The reduction of angiogenesis has been shown in an experimental model of human colon cancer, where tinzaparin administration 24 h after angiogenesis stimulation by VEGF led to a reduction of the angiogenic index to the control level [39]. Tinzaparin exerts its anti-neoangiogenic activity through induction of greater levels of production of tissue factor pathway inhibitor (TFPI) by epithelial cells than any other low molecular weight heparin, thus inhibiting tissue factor (TF) and consequently VEGFR. However, LMWHs are known to inhibit heparanase, an endoglycosidase rarely expressed in normal tissue but overexpressed in pancreatic tumors. The activity of LMWHs is demonstrated through the inhibition of VEGF-A and FGF-2 as well as the increased release of TFPI, suggesting an antagonistic role of LMWHs in angiogenesis due to heparanase. Vascular endothelial growth factor A (VEGF-A) binding to the receptor tyrosine kinase VEGFR2 triggers multiple signal transduction pathways, which regulate endothelial cell responses that control vascular development [40].

Τhere are several reports about the role of tinzaparin in various types of cancer, where it appears to attenuate tumor growth but -most importantly- inhibit the metastatic cascade process. Using a mouse pancreatic cancer cell line injected into mouse models, tinzaparin was able to impede tumor growth without affecting the thrombotic phenotype [41]. Another study demonstrated tinzaparin’s ability to upregulate the expression of E-cadherin in pancreatic tumor cells, a marker of decreased disseminating capacity and reduced metastatic potential [42]. Tinzaparin administration in mice injected with melanoma cells was capable of diminishing the metastatic potential to distant organs by disturbing the P-, L-selectin and VLA-4/VCAM-1 interconnections. In addition, tinzaparin suppressed the binding of malignant cells expressing the C-X-C chemokine receptor type 4- to their specific ligand on normal tissue, resulting in a substantial reduction of the dissemination of human breast cancer cells to the lung [43]. The antimetastatic properties of tinzaparin have also been observed in a B16 melanoma cell lung metastasis model in mice [15]. Administration of tinzaparin prior to the injection of melanoma cells resulted in a decrease in lung tumor formation by 89% compared with controls, whereas daily administration of tinzaparin for two consecutive weeks reached an additional reduction in lung tumor formation, reaching almost 96% [44].

Our results demonstrate that tinzaparin reduces the expression of VEGFR2 resulting in lower vessel formation in the tumor, which results in decreased nutrient supply. VEGFR2 activation stimulates cell proliferation and survival, activities that were shown to be significantly attenuated in our experiments. We showed an important down-regulation of VEGFR2 and suppressed cell proliferation, observed with PCNA marker staining. Our data also demonstrated a reduction in survival of pancreatic cancer cells through the mechanism of apoptosis, as seen by in vitro and in vivo experiments using caspase-3.

Summarizing the above, we conclude that the triple-drug combination of chemotherapeutic drugs and tinzaparin is capable of reducing significant tumor growth and development. This substantial effect is the result of the orchestrated action of multiple pathways that affect cellular proliferation, apoptosis and neoangiogenesis. Our results highlight the antitumor activity of tinzaparin when it is combined with nab-paclitaxel and gemcitabine in in vitro and in vivo experiments in PC models according to a mutant KRAS profile. It should be noted here that the dose of tinzaparin administered (250 UI/Kg) to the mice was the highest possible without bleeding, so it can be considered hyper-prophylactic as used in the PaCT study [41]. A clinical study using tinzaparin in patients with non-small cell lung cancer did not have the expected PFS results, probably because the tinzaparin dose was 100 UI/Kg, lower than the 175 UI/Kg used in the PaCT study [45]. Our results are in absolute correspondence with studies such as PACT where administration of tinzaparin during chemotherapy increased progression-free survival (PFS).

Tinzaparin possibly has similar synergistic potential when combined with other chemotherapeutic regimens used in PC, such as FOLFIRINOX. This is due to the fact that the pharmacological action of nab-paclitaxel, plus gemcitabine and (5-FU), irinotecan, and oxaliplatin—which comprise the formulation called FOLFIRINOX- is based on targeting DNA synthesis (except folinic acid) [46,47,48].

PaCT (Pancreatic cancer & tinzaparin) is a retrospective observational study that collects data regarding progression-free survival (PFS) in advanced or metastatic PC patients who received thromboprophylaxis with tinzaparin during chemotherapy with nab-paclitaxel and gemcitabine. In this study, the median PFS was 7.9 months. Out of 14 similar studies (involving 2994 patients), identified via a systematic search, it was settled that the weighted PFS of patients receiving Nab-P and G but no anticoagulation was 5.6 months. Therefore, patients receiving tinzaparin had 39.54% higher PFS than patients without thromboprophylaxis (*p* < 0.05). Tinzaparin was administrated in a “hyper-prophylactic” 10.000 Anti-Xa IU (175 UI/Kg) or even higher full treatment doses. Possible reasons for the increase in PFS were the treatment of venous thromboembolism, a prevalent condition in pancreatic cancer, and the possible antitumor role of tinzaparin [49].

## 4. Materials and Methods

### 4.1. Cell Lines

Human pancreatic cancer cell lines, PANC-1 and MIA PaCa-2, were cultured according to the guidelines of the American Type Culture Collection (ATCC). More specifically, cell lines were cultured in Dulbecco’s modified Eagle medium (Thermo Fisher Scientific, Waltham, MA, USA) supplemented with 10% FBS, 100 U/mL penicillin-streptomycin and 2 mM L-glutamine.

### 4.2. Cell Viability—XTT Assay

The assessment of pancreatic cancer cell proliferation was performed with the XTT Cell Proliferation Assay Kit (10010200, Cayman Chemical, Ann Arbor, MI, USA). Cells were seeded in a 96-well plate at a density of 103–105 cells/well in the aforementioned medium and cultured in a 100-μL medium with or without the tested drugs (1 μΜ gemcitabine, 1 μΜ nab-paclitaxel, 2 U/mL tinzaparin) in a CO_2_ incubator at 37 °C for 48 h. Afterward, 10 μL of XTT mixture was added to each well and was mixed gently for 1 min on an orbital shaker. The plates were incubated for 2 h at 37 °C in a CO_2_ incubator and the absorbance of each sample was measured using a microplate reader at 450 nm.

### 4.3. Western Blotting

RIPA buffer was used for the preparation of whole-cell lysates. The protein concentration was determined using the Bradford method (5000006, Bio-Rad Laboratories, Hercules, CA, USA). A total of 25 μg of protein was resolved on SDS-PAGE and transferred to nitrocellulose membranes (Whatman, Scheicher & Schuell, Dassel, Germany). Membranes were incubated with primary antibodies overnight at 4 °C. After incubation time, membranes were washed with TBS-T and incubated with the appropriate secondary antibody, for 1 h at RT. Antibodies were used against VEGFR #9698, cleaved caspase-3 #9661 and PARP-1 #9542 (Cell Signaling Technology, Danvers MA, USA) and Actin (sc-8035, Santa Cruz Biotechnology, Dallas, TX, USA). After incubation with HRP-conjugated secondary antibodies, detection of immunoreactive bands was performed with the Clarity Western ECL Substrate (Bio-Rad, Laboratories, Hercules, CA, USA). Relative protein amounts were evaluated by a densitometry analysis using ImageJ software (La Jolla, CA, USA) and normalized to the corresponding actin levels.

### 4.4. In Vivo Experiments

All in vivo experiments were performed on NOD/SCID mice between 6 and 8 weeks of age. All procedures were carried out in accordance with the guidelines for animal experimentation following the National and Kapodistrian University of Athens Medical School Bioethics Committee in agreement with the European Union (approval no. 3233/26-06-2018). PANC-1 cell suspensions (1 × 10^6^) in 100 μL PBS were injected into the right flank of each mouse and allowed to grow for approximately 2 weeks to the point that they were palpable. The mice were randomly divided into groups (*n* = 5 per group) for each treatment (control, tinzaparin, nab-paclitaxel, nab-paclitaxel + tinzaparin, gemcitabine, gemcitabine + tinzaparin, nab-paclitaxel + gemcitabine, nab-paclitaxel + gemcitabine + tinzaparin). We used tinzaparin at 10 mg/Kg administered daily by subcutaneous injection (s.c.), nab-paclitaxel at 25 mg/Kg and gemcitabine at 60 mg/Kg twice per week by intraperitoneal injection (i.p.). The mice were euthanized, and tumors were measured and excised after 15 days of treatment. Tumor volume was calculated using the following formula: 1/2(length × width^2^).

### 4.5. Immunohistochemistry

Tumors from NOD-SCID mouse xenografts were fixed in 10% formalin solution and embedded in paraffin for sectioning at 4 μm. Sections were then deparaffinized in xylene and dehydrated through a graded ethanol series. Antigen retrieval was performed by heating samples for 20 min at 95 °C in citrate buffer (pH 6.0), and endogenous peroxidase was blocked with 3% hydrogen peroxide for 10 min at room temperature (RT). Afterward, sections were washed with PBS and blocked with Normal Goat Serum 5% (NGS) for 1 h. Sections were then incubated with cleaved caspase-3 antibody (rabbit monoclonal antibody, #9661, Cell Signaling Technology, Danvers, MA, USA), VEGFR-2 antibody (rabbit monoclonal antibody, #9698, Cell Signaling Technology, Danvers, MA, USA) at a dilution of 1:400 at 4 °C overnight. After washing with PBS, sections were incubated with biotinylated secondary antibodies (cat. no. 20775; Merck Millipore, Burlington, MA, USA) for 10 min at RT. Following that, sections were incubated with Streptavidin HRP (cat. no. 20774; Merck Millipore, Burlington, MA, USA for 10 min at RT, and the reaction was visualized using 3,3′-diaminobenzidine. Finally, specimens were counterstained with Mayer’s hematoxylin at RT for 1 min. Images were captured with a Nikon Eclipse 80i microscope with digital camera image system (Cellsens). Samples were blindly inspected by an experienced pathologist.

### 4.6. Immunofluorescence

Initially, sections were incubated with citrate buffer (pH 6.0) for antigen retrieval and then blocked in 5% NGS for 1 h and incubated at a dilution of 1:400 at 4 °C overnight with PCNA antibody (mouse monoclonal antibody, #2586, Cell Signaling Technology, Danvers, MA, USA). After washing with PBS, sections were incubated with Alexa Fluor 568 secondary antibody (1:500, Thermo Fisher Scientific, Waltham, MA, USA). Sections were examined using an Olympus FV1000 confocal microscope.

### 4.7. Statistical Analysis

Statistical significance was analyzed by the two-tailed Student t-test. Values of *p* < 0.05 were considered to represent statistically significant group differences. All data represent mean ± SD. Microsoft Excel (Office 2017, Microsoft, Redmond, WA, USA) was used. We also used Graph-Pad Prism (V5.0., San Diego, CA, USA and Gpower V3.1 (University Düsseldorf, Germany) to calculate with accuracy the sample size of experimental animals.

## 5. Conclusions

Pancreatic cancer remains a considerable challenge in terms of its treatment. Delayed diagnosis, TME, tolerance to drugs such as gemcitabine are some of the causes of low survival of PC patients. Although life expectancy has increased in recent years with the administration of drug combinations, further improvement is required in order to find more effective treatments. Based on the results of our study, we believe that the administration of tinzaparin in patients with pancreatic cancer who carry the KRAS mutation is a step towards this direction. More specifically, co-administration with nab-paclitaxel and gemcitabine in NOD/SCID mice shows further tumor reduction through apoptosis, as well as a decrease of tumor vascularity that can possibly result in alleviating desmoplasia and lead to increased penetration of chemotherapeutic drugs. Further prospective randomized research with various dose concentrations above 150 UI/Kg is recommended in order to determine whether tinzaparin could potentially provide a valuable synergistic effect to conventional therapeutic compounds.

## Figures and Tables

**Figure 1 ijms-22-07053-f001:**
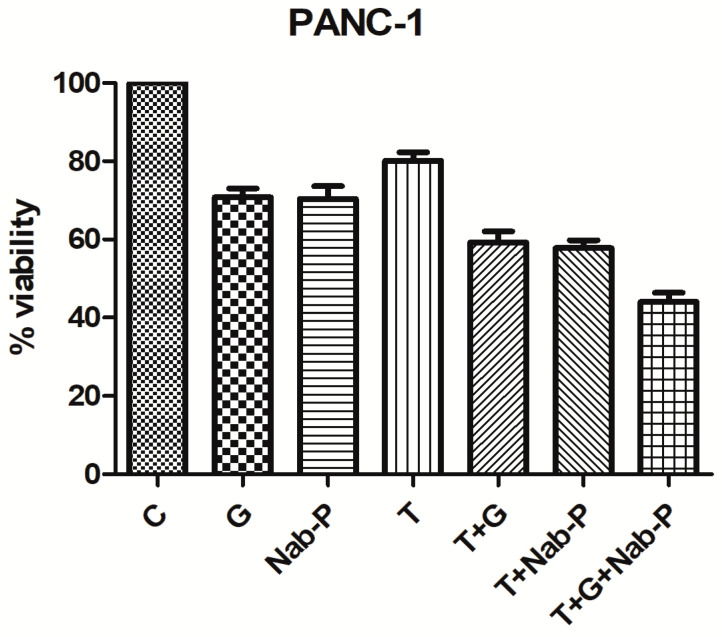
Tinzaparin administration with drug combinations results in further decrease in cell viability. Cell viability with XTT assay of the mutant KRAS pancreatic cancer cell lines PANC-1 and MIAPaCa-2 after 48 h treatment with 1 μM Nab-paclitaxel (Nab-P), 1 μM gemcitabine (G) and 2 UI/mL tinzaparin (T) alone or in combination. All data represent mean ± SD and three independent experiments.

**Figure 2 ijms-22-07053-f002:**
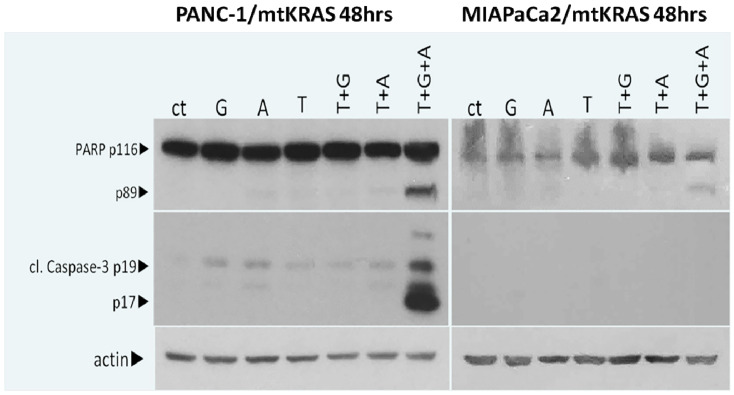
Tinzaparin administration with drug combinations leads to an additional increase of apoptosis. Western blot analysis of pancreatic cancer cell lines PANC-1 and MIAPaCa-2 after 48 h treatment with 1 μM nab-paclitaxel (Nab-P), 1 μM gemcitabine (G) and 2 UI/mL tinzaparin (T) alone or in combination. Protein levels of apoptotic cell death were identified by antibodies against PARP and cleaved caspase-3. Protein levels were normalized against actin. A representative image of two replicates is shown.

**Figure 3 ijms-22-07053-f003:**
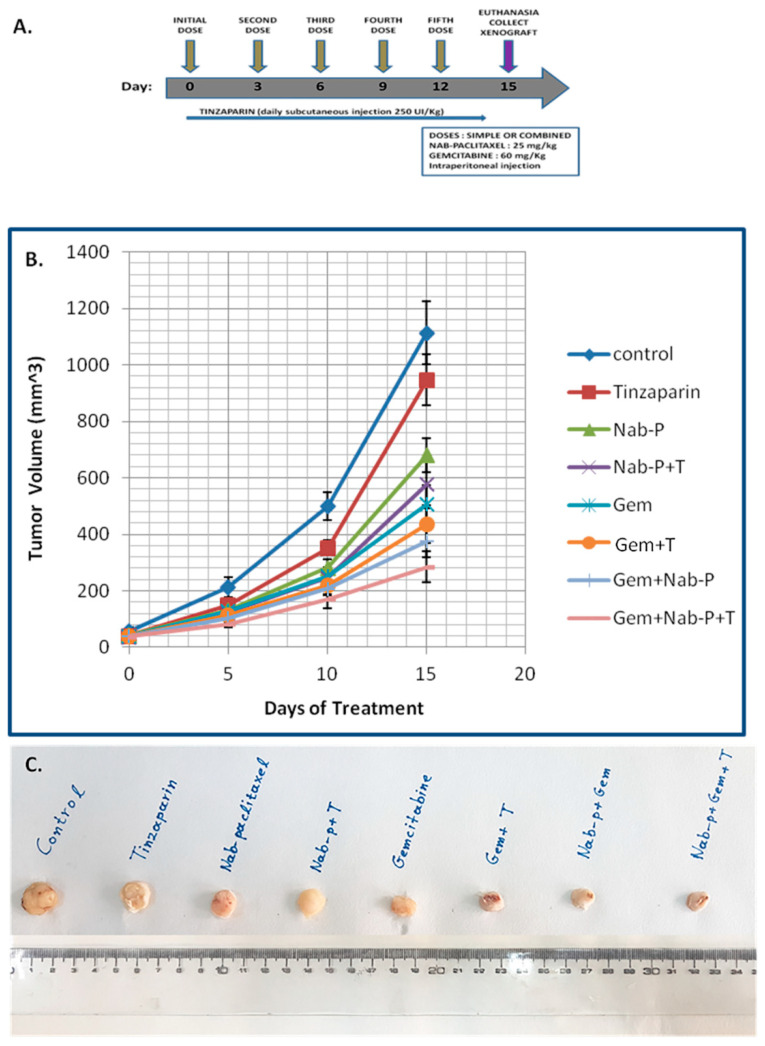
Tinzaparin results in tumor growth inhibition in combination with chemotherapeutic drugs in pancreatic tumor xenografts. (**A**) Mice were injected subcutaneously in the right flank with 0.1 mL PBS containing 10^6^ PANC-1 human pancreatic cancer cells. After 15 days of treatment, mice were euthanized and tumors were collected for further analysis. (**Β**) The diagram shows the tumor volume for each group (control, tinzaparin, nab-paclitaxel, nab-paclitaxel + tinzaparin, gemcitabine, gemcitabine + tinzaparin, nab-paclitaxel + gemcitabine, nab-paclitaxel + gemcitabine + tinzaparin) during treatment. All data represent mean ± SD, (*n* = 5/group) (**C**) Representative tumors for each group with different treatments or no treatment (control).

**Figure 4 ijms-22-07053-f004:**
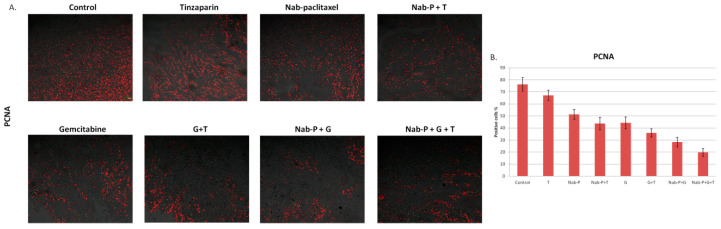
Tinzaparin leads to reduced proliferation when used in combination with chemotherapeutic drugs. (**A**) Immunofluorescence with PCNA antibody to 4 μm paraffin sections. Representative images (PCNA-red spots and tissue-bright field) of each group (control, tinzaparin, nab-paclitaxel, nab-paclitaxel + tinzaparin, gemcitabine, gemcitabine + tinzaparin, nab-paclitaxel + gemcitabine, nab-paclitaxel + gemcitabine + tinzaparin) at 100× magnification, scale bar = 100 μm. (**B**) Diagram showing the percentage of positive stained cells. *n* = 10/group. All data represent mean ± SD.

**Figure 5 ijms-22-07053-f005:**
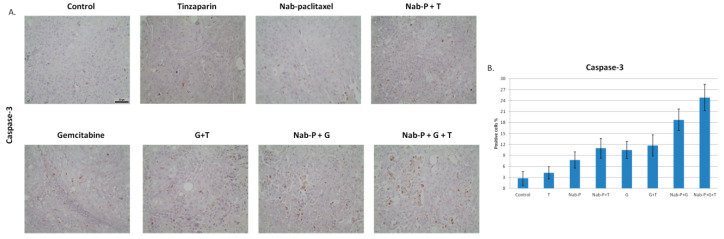
Tinzaparin induces apoptosis when administered as part of conventional double or triple drug regimes. (**A**) Immunohistochemistry with anti-caspase-3 antibody at 4 μm paraffin sections. Representative photos of each group (control, tinzaparin, nab-paclitaxel, nab-paclitaxel + tinzaparin, gemcitabine, gemcitabine + tinzaparin, nab-paclitaxel + gemcitabine, nab-paclitaxel + gemcitabine + tinzaparin) 200× magnification, scale bar = 50 μm. (**B**) The diagram shows the percentage of positive cells for caspase-3. *n* = 10/group. All data represent mean ± SD.

**Figure 6 ijms-22-07053-f006:**
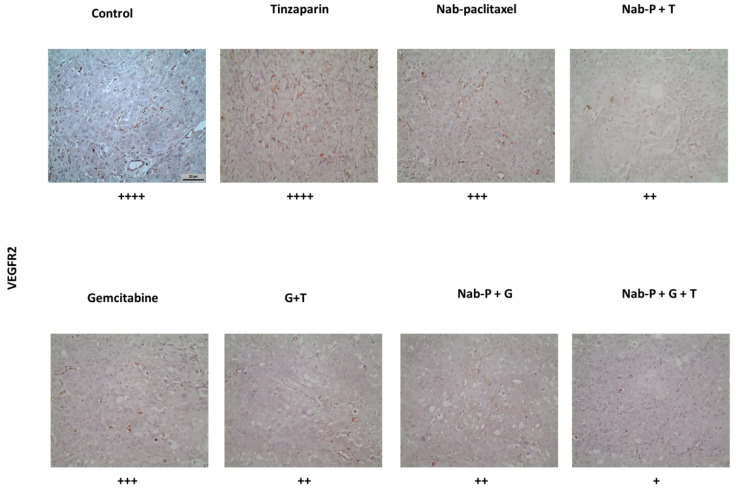
Tinzaparin augments further decrease in neoangiogenesis when added to double or triple combinations. Representative photos of each group (control, tinzaparin, nab-paclitaxel, nab-paclitaxel + tinzaparin, gemcitabine, gemcitabine + tinzaparin, nab-paclitaxel + gemcitabine, nab-paclitaxel + gemcitabine + tinzaparin) after staining with anti-VEGFR2 at paraffin sections. Symbols are denoted as: + low expression, ++ mid expression, +++ high expression, ++++ abundant expression. *n* = 5/group, 200× magnification, scale bar = 50 μm.

## Data Availability

The data presented in this study are available on request from the corresponding authors.

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
