# Peer review of "Combinatorial Treatment of Tinzaparin and Chemotherapy Can Induce a Significant Antitumor Effect in Pancreatic Cancer"

_ijms, 2021, doi:10.3390/ijms22137053_

Round 1
Reviewer 1 Report
In this manuscript entitled “Combinatorial treatment of Tinzaparin and chemotherapy can induce a significant antitumor effect in pancreatic cancer”, Panagiotis Sarantis et al., describe the use of Tinzaparin, a low molecular weight heparin, in order to potentiate the pro-apoptotic effect of Nab-paclitaxel/Gemcitabine in pancreatic cancer treatments.
This topic is of true interest for the scientific community and the manuscript is clear. However, some points remain to be verified in order to clarify the scientific message of this study.
Comments:
1/ English needs to be verified.
2/ In several figures there is no statistical analyses. We have no information on how many times the experiment was performed neither the statistical analyses of the claimed difference. This comment is specific to Figures 1, 2 ,3 and 6.
3/ Regarding Figure 1, why the viability measurement was performed at 48 hours? Do the authors checked longer time points in order to verify if there are no further decreased survival in those?
4/ Figure 2. There is no analyses or quantification of this western blot. Is this Western blot representative of several experiments? Authors should show a quantification of several western blot and show a representative picture. Why is there no PARP cleavage in single or double treatments? A total caspase 3 (non-cleaved) would be necessary to interpret those western blots.
5/ Figure 3. There are no statistical analyses of this figure. Are the differences induced by Tinzaparin in the various combination significant?
6/ Figure 6. The authors should quantify those differences by analyzing numerous pictures of each conditions are report this on a histogram with statistical variation and analyses.
7/ In conclusion the authors could mention if Tinzaparin could potentiate other chemotherapeutic protocols used against PC as Folfirinox.
Author Response
In this manuscript entitled “Combinatorial treatment of Tinzaparin and chemotherapy can induce a significant antitumor effect in pancreatic cancer”, Panagiotis Sarantis et al., describe the use of Tinzaparin, a low molecular weight heparin, in order to potentiate the pro-apoptotic effect of Nab-paclitaxel/Gemcitabine in pancreatic cancer treatments.
This topic is of true interest for the scientific community and the manuscript is clear. However, some points remain to be verified in order to clarify the scientific message of this study.
AUTHOR RESPONSE: Initially, we would like to thank the reviewer for the detailed remarks and fruitful comments that will be beneficial for the current manuscript. Below we provide our detailed answers to each comment one-by-one to ensure the clarity of our statements
Comments:
1/ English needs to be verified.
AUTHOR RESPONSE: We have made the necessary corrections in order to improve the language of the manuscript
2/ In several figures there is no statistical analyses. We have no information on how many times the experiment was performed neither the statistical analyses of the claimed difference. This comment is specific to Figures 1, 2 ,3 and 6.
AUTHOR RESPONSE: We thank the reviewer for this remark. We have added the statistical analyses at the figures that were missing and the number of experiments performed (added at figure legends). The analyses were focused on the basic goal of the current study, which was to emphasize the differences between the triple combination compared to the control and the double combination. As for figure 3, the experiment was performed once due to limitations derived from bioethical rules and regulations (as every animal experiment).We used the exact number of mice recommended by the use of Gpower analysis programme (5 for each group). As for figure 6, there are 5 photos from each group from the aforementioned mice.
3/ Regarding Figure 1, why the viability measurement was performed at 48 hours? Do the authors checked longer time points in order to verify if there are no further decreased survival in those?
AUTHOR RESPONSE: We thank the reviewer. We checked the viability of cells in 24 and 72 hours. In 24 hours the treatment did not show a significant cells effect on the cells. At 48 hours incubation, the treatment exhibited substantial effects to their survival, while treatment for 72 hours was almost the same as 48 hours. This is the reason why we chose to present the 48 hours-treatment.
4/ Figure 2. There is no analyses or quantification of this western blot. Is this Western blot representative of several experiments? Authors should show a quantification of several western blot and show a representative picture. Why is there no PARP cleavage in single or double treatments? A total caspase 3 (non-cleaved) would be necessary to interpret those western blots
.
AUTHOR RESPONSE: We thank the reviewer for the remark.
- The data produced with a western blot are typically considered to be semi-quantitative, western blot technique provides a relative comparison of protein levels, not an absolute measure of the quantity. Furthermore, in Figure 2 we present the protein levels of PARP and Caspase-3. The present of cleavage of these two proteins identify the apoptotic cell death, qualitative analysis is not required.
- Figure 2 is the representative picture of 2 individual experiments.
- The present of PARP cleavage identify the present of apoptotic cell death. Thus, in single and double treatment, the absence of PARP cleavage means the absence of apoptotic cell death. Nonetheless in Nab-P and in T+G/T+Nab-P treatment, a slight PARP cleavage is detected.
- Cleaved Caspase-3 (Asp175) Antibody (Cell signaling #9661) detects endogenous levels of the large fragment (17/19 kDa) of activated caspase-3 resulting from cleavage adjacent to Asp175. This antibody does not recognize full length caspase-3 or other cleaved caspases. Protein levels were normalized against actin.
5/ Figure 3. There are no statistical analyses of this figure. Are the differences induced by Tinzaparin in the various combination significant?
AUTHOR RESPONSE: We thank the reviewer for the remark. Ιt was our omission not to calculate the p value of the difference in tumor volume between double (Nab-P+G) and triple combination. There is a statistical significance of p=0.0329 between them, a remark that has been embodied in the manuscript. The significance among the triple and the rest of combinations is far more than 0.0329.
6/ Figure 6. The authors should quantify those differences by analyzing numerous pictures of each conditions are report this on a histogram with statistical variation and analyses.
AUTHOR RESPONSE: We thank the reviewer for the comment. At figures 4 and 5, we performed the analysis you recommended as it was possible to measure the percentage of positively stained cells compared to their total number. Ιn this case, this was not feasible therefore we prefer to present a qualitative analysis of the photos, something that is very common in bibliography for VEGFR-2
7/ In conclusion the authors could mention if Tinzaparin could potentiate other chemotherapeutic protocols used against PC as Folfirinox
AUTHOR RESPONSE: We thank the reviewer for the remark. We have added the above text in the manuscript:
“Tinzaparin possibly has similar synergistic potential when combined with other chemotherapeutic regimens used in PC, such as Folfirinox. This is due to the fact that the pharmacological action of Nab-paclitaxel,plus Gemcitabine and (5-FU), irinotecan, and oxaliplatin - which comprise the formulation called Folfirinox- is based on targeting DNA synthesis (except folinic acid) [46-48].”
Reviewer 2 Report
The authors provides an interesting model for combining drug treatment of Tinzaparin and two traditional chemotherapy drugs on pancreatic cancer model. We look forward further clinical trials for more efficient tumor killing effect in pancreatic and other cancer patients.
Major revision :
1. The author should describe how to decide the concentration of in vitro Tinzaparin treatment since this drug is used to treat coagulation and itself should not cause the cell death.
2. Figure 1 should be improved by removing the shadow and the resolution. The position of error bars is not precise. Please refrain from manually drawing the error bar on the bar chart.
3. The title of 2.2 need to be re-written. Line 122.
4. Figure2, the title of figure show 24 hrs, however, the figure legends show 48 hrs
5. In figure 1 and 2, the combination of G+A is missing compared to later in vivo treatment.
6. In figure 3c, difference between G+A and T+G+A may not be significant. The authors are suggested to provide raw images of a table describing the volume of all dissected tumor samples to solidify their observation.
Minor revision:
1. Line 58 m square?
2. The title of 2.3 should be NOD/SCID Line140
3. Line 146, Panc-1 or PANC-1 should be consistent.
4. Why MIAPACA2 shows no apparent apoptotic markers? The authors should provide further evidences by IFA.
5. Whether tinzaparin combanation treatment can lower the dosage of G+A is an interesting question.
6. Figure 3B, GEM or Gem should be consistent.
7. Line 347, 1x10^6 should be reformatted.
Author Response
The authors provides an interesting model for combining drug treatment of Tinzaparin and two traditional chemotherapy drugs on pancreatic cancer model. We look forward further clinical trials for more efficient tumor killing effect in pancreatic and other cancer patients.
AUTHOR RESPONSE: We would like to thank the reviewer for the apt comments and constructive remarks. Below we provide our detailed answers to each comment one-by-one to ensure the clarity of our statements
Major revision :
The author should describe how to decide the concentration of in vitro Tinzaparin treatment since this drug is used to treat coagulation and itself should not cause the cell death.
AUTHOR RESPONSE: We thank the reviewer. In order to evaluate the effect of tinzaparine on cells we tried 3 different concentration of this drug on pancreatic cancer (PC) cells. In preliminary data, the higher dose of tinzaparine appeared to decrease the protein levels of VEGFR in two PC cell lines (PANC-1 and MIAPaCa-2). Thus, we continued with the high dose of 250 UI/Kg for in vivo experiments.
- Figure 1 should be improved by removing the shadow and the resolution. The position of error bars is not precise. Please refrain from manually drawing the error bar on the bar chart.
AUTHOR RESPONSE: We thank the reviewer for the remark.
We have reformed figure 1 with the help of GraphPad Prism in order to resolve the issues you have reported.
- The title of 2.2 need to be re-written. Line 122.
AUTHOR RESPONSE: We thank the reviewer for the comment. We rewrote it and hope it's ok now.
- Figure2, the title of figure show 24 hrs, however, the figure legends show 48 hrs
AUTHOR RESPONSE: We thank the reviewer for his observeness and the fruitful remark. We have made the correction and we sincerely apologize for this mistake.
- In figure 1 and 2, the combination of G+A is missing compared to later in vivo treatment.
AUTHOR RESPONSE: We thank the reviewer for the remark. In in vitro experiments we tried to evaluate the effect of tinzaparin on cells. Thus, we combined tinzaparin with two drugs (Νab-P and gemcitabin) in order to evaluate the effect of the combinations. We continued, with the more significant for tinzaparin treatment and its evaluation, in vivo experiments, with all combinatorial treatments including the double combination of G+A.
To further verify the impressive in vitro results we obtained with Tinzaparin, we proceeded to in vivo experiments with all the combinatorial treatments including the double combination of G+A.
- In figure 3c, difference between G+A and T+G+A may not be significant. The authors are suggested to provide raw images of a table describing the volume of all dissected tumor samples to solidify their observation.
AUTHOR RESPONSE: We thank the reviewer for the comment. There is a statistical significance of p=0.0329 between double (Nab-P+G) and triple combination (added in the manuscript). Each group contained 5 animals where one was selected randomly to be the representative for each treatment. It is very common in bibliography in experiments with many groups of mice, to present only one tumor from each treatment group.
The total number of mice used was calculated with GPower Analysis Programme, which measures the number of animals required to have a significant result.
Minor revision:
Line 58 m square?
AUTHOR RESPONSE: Corrected
- The title of 2.3 should be NOD/SCID Line140
AUTHOR RESPONSE: Thank you for the comment. If we have understood correctly, you meant that the word xenograft before NOD/SCID is not right, therefore we have deleted it.
- Line 146, Panc-1 or PANC-1 should be consistent.
AUTHOR RESPONSE: Corrected
- Why MIAPACA2 shows no apparent apoptotic markers? The authors should provide further evidences by IFA.
AUTHOR RESPONSE: Thank you for the apt comment. MIAPaCa-2 is a very aggressive cell line compared to PANC-1 and shows no apaparent apoptotic markers, probably because it is resistant to this treatment or because the specific cancer cells die through a different mechanism. Cancer cell death might come as a result of autophagy or necraptosis but not from apoptotic caspase cascade. Despite the great interest of this question, the molecular mechanism is beyond the analysis of the current manuscript.
- Whether tinzaparin combanation treatment can lower the dosage of G+A is an interesting question.
AUTHOR RESPONSE: Thank you for the comment, indeed that is a very interesting question. Unfortunately, this can be assessed to the current manuscript, especially the in vivo experimental part. However, based on our results we can deduce that –as there were no side effects with the dosing that was used- the tumor growth inhibition was significant and probably with lower doses we would have a smaller reduction in tumor volume.
Figure 3B, GEM or Gem should be consistent.
AUTHOR RESPONSE: Corected
Line 347, 1x10^6 should be reformatted.
AUTHOR RESPONSE: Reformatted
Reviewer 3 Report
Combinatorial Treatment of Tinzaparin and Chemotherapy Can Induce a Significant Antitumor Effect in Pancreatic Cancer
I would like to congratulate the authors with a nicely designed and methodically executed study shedding some new light on the effect of LMWH (or maybe just Tinzaparin?) in combined systemic treatment of pancreatic cancer. However I have a identified a few issues having to be addressed.
Major comments:
The data on numbers of experimental animals in groups with different treatments neither in the “Methods” nor in the Results sections. Similarly there is no data on sample size calculations. Consequently, the statistical significance of obtained results becomes doubtful.
There are too many explanations and speculations within the Results section that might be more appropriate in the Discussion section. The Results should be presented in a concise form.
The Conclusions have to be updated to exclusively answer the objectives of the study and avoiding any speculations.
Minor comments
The introduction section is too extensive, excessively detailed.
The English language needs editing.
Author Response
I would like to congratulate the authors with a nicely designed and methodically executed study shedding some new light on the effect of LMWH (or maybe just Tinzaparin?) in combined systemic treatment of pancreatic cancer. However I have a identified a few issues having to be addressed.
AUTHOR RESPONSE: Initially, we would like to thank the reviewer for his constructive comments that aim to the improvement of our manuscript.
Major comments:
The data on numbers of experimental animals in groups with different treatments neither in the “Methods” nor in the Results sections. Similarly there is no data on sample size calculations. Consequently, the statistical significance of obtained results becomes doubtful.
AUTHOR RESPONSE: We thank the reviewer for the remark. We have reported the exact number of animals that were used for in vivo experiments (5 per group) at the Materials and Methods section. We have added the statistical analyses that were missing and also embodied the number of experiments performed (added at Figure Legends). The analyses were focused on the main goal of the current study, the comparison of tumor growth inhibition between a triple drug combination and the untreated/ double combination
There are too many explanations and speculations within the Results section that might be more appropriate in the Discussion section. The Results should be presented in a concise form.
AUTHOR RESPONSE: We thank the reviewer for the comment. We are in agreement with this fruitful remark, as there was extensive analysis at the Result section, therefore we have made the corrections.
The Conclusions have to be updated to exclusively answer the objectives of the study and avoiding any speculations.
AUTHOR RESPONSE: We thank the reviewer for the comment. We have made many changes towards the proposed direction and we sincerely hope we have covered your apt remarks to a great extent.
Minor comments
The introduction section is too extensive, excessively detailed.
AUTHOR RESPONSE: We thank the reviewer for the remark. We deleted 2 unnecessary sentences.
The English language needs editing.
AUTHOR RESPONSE: We have made the necessary corrections in order to improve the language of the manuscript
Round 2
Reviewer 1 Report
No specific suggestion